# Reducing Surface Roughness of 3D Printed Short-Carbon Fiber Reinforced Composites

**DOI:** 10.3390/ma15207398

**Published:** 2022-10-21

**Authors:** Raluca Maier, Sebastian-Gabriel Bucaciuc, Andrei Cristian Mandoc

**Affiliations:** Composite Materials Laboratory for Aeronautical Field, Romanian Research & Development Institute for Gas Turbines—COMOTI, 220D Iuliu Maniu Av., 061126 Bucharest, Romania

**Keywords:** 3D printing, surface roughness, post-processing, laser polishing

## Abstract

A 100 W fibre laser source was used to minimize the surface roughness of 3D-printed Onyx parts. Furthermore, this study aimed to determine the mechanism of surface finishing, the influence of the laser process parameters (laser power, pulse frequency, and laser scanning path) on the surface morphology, and the influence of the scanning path on the dimensional accuracy of the investigated Onyx 3D-printed specimens. A significant reduction in surface roughness of 91.15% was achieved on the S3 Onyx 3D-printed specimen following laser surface polishing treatment using a 50 W laser power and a frequency of 50 kHz. The laser scanning path had little influence on the surface roughness, but had a major impact on the geometrical deviation of the treated sample.

## 1. Introduction

Nowadays, additive manufacturing (AM) technology has gained serious attention thanks to its major advantages such as product development, prototype manufacturing, and so on. It is also capable of overcoming the conventional disadvantages encountered in other manufacturing techniques, becoming the most efficient approach towards product development. Fabricating custom products is a major challenge considering market requests like low-cost and rapid manufacturing. AM, known as 3D printing, uses a layer-by-layer manufacturing technique. Unlike conventional methods, this technology provides a good reduction in waste materials and is also capable of shortening the manufacturing time, reducing the cost of geometrically complex parts in a relatively short period of time. Three-dimensional (3D) printing of polymer composites with enhanced mechanical properties solves the previous limitations by the addition of reinforcements, such as particles, fibres, or nanomaterials, into thermoplastic polymers, allowing the fabrication of polymer matrix composites, which are characterized by high performance and excellent functionality [1,2].

The whole process consists of five steps: (1) CAD design for 3D printing, (2) export to STL file, (3) parameter setting, (4) printing, and (5) surface finishing and/or post-processing techniques.

These composites show poor mechanical properties compared with composites manufactured by conventional methods, because composites reinforced with short fibres or particles are mechanically inferior to composites reinforced with continuous fibres. The possibility of employing continuous fibre reinforced thermoplastic composites may lead to a product with much higher mechanical performance, which could potentially be useful for advanced applications [3]. Among a broad spectrum of AM processes [4] for polymer and composite materials, FDM (fused deposition modelling) is the first choice owing to its flexibility, complex shapes’ development, higher printing speed, low cost, high strength and toughness, non-toxicity, and the diversity of materials in comparison with other AM technologies. A further later development of FDM is continuous filament fabrication (CFF) from Markforged [5,6]. In addition to printing thermoplastics, Markforged also adapts the FFF (fused filament fabrication) process to print non-plastics. In CFF manufacturing, a FFF printer with a second nozzle places continuous carbon fibres, fiberglass, or Kevlar^®^ in one piece, but instead of melting the entire filament, the heat of its nozzle is used to “integrate” into the thermoplastic layer. CFF is a promising alternative to conventional processes for the fabrication of CFRTPCs (continuous fiber-reinforced thermoplastic composites), such as vacuum forming, filament winding, pultrusion, bladder-assisted molding, or compression, which require expensive facilities and equipment, such as autoclaves or complex rigid molds for out-of-autoclave processes, hindering the wide application of composites. Although this technology is still in its infancy, two methods have been reported for embedding continuous fibres into the thermoplastic filament: embedding the continuous fibre in the injector in a “co-extrusion” process [2] or embedding the continuous fibre directly in the component with a dual extrusion method. MarkForged^®^ has successful implemented the second method, claiming a remarkable improvement in the mechanical performances of 3D-printed composites [7].

FDM is a complex process with many parameters that influence product quality and material properties, and the combination of these parameters is often difficult to understand; however, it is clear that the staircase effect is one of the key characteristics of all AM manufactured parts. This is the main factor that has a direct influence on surface roughness and can be observed on micro and macro levels. Many authors focused their attention on optimizing 3D printing parameters, early in the stage of pre-processing, for achieving a fine surface, but the printing time was high owing to the very small thickness of the layer.

The main issue in additive manufacturing refers to the quality of the surfaces. It is essential to maintain the long-term performance of the printed parts, as well as their functionality. Normally, in the composite manufacturing process, the surface finishing stage cannot be skipped, with dimensional accuracy playing an important role in achieving the desired surface. Surface roughness has a major influence upon the mechanical behaviour of the printed parts, like crack initiation, wear resistance, fatigue life, and so on. There are two main strategies that can be used for reducing the roughness of the 3D-printed parts: pre-processing and post-processing. The pre-processing stage is given by specific parameters of the printing machine, part orientation, geometry, and so on. Post-processing can be divided into two phases: primary and secondary. Primary post-processing includes the necessary steps to follow for all 3D-printed parts to be usable. Secondary post-processing includes surface finishing, a process that improves the properties, performance/functions, or even the aesthetics of the parts. The need for a reduction in surface roughness will be defined in compliance with the experimental approach/purpose for which the parts were printed. In this investigation, the secondary post-processing phase is important because it refers strictly to improving surface roughness via laser polishing.

Many researchers have had different perspectives on improving the quality of surfaces. Currently, the most common techniques are as follows:Sanding: performed manually, and thus unsuitable for industrial production [8].Chemical post-processing: chemical baths can be used to smooth printed parts via ME (material extrusion) [9].Epoxy coating: applying homogenized epoxy resin and hardener, which improves the surface roughness of the printed part [10].

None of the mentioned techniques above are used at an industrial scale. Therefore, a new technique suitable for large-scale applications is required. To bring CFF to an industrial level, an auxiliary process that is automatable, adaptable, and does not affect mechanical properties of the part is necessary. To this end, auxiliary assistance after the part is printed can be utilized. The application of heat on the rough surface can be an easy way to achieve a smooth surface; however, it is difficult to control a temperature field to reach the desired results. It is challenging to reduce the surface roughness to desired values while not affecting sub-surface features. Therefore, the use of focused energy sources instead of energy fields is a necessity in making sure that only the desired geometry is removed [11] as shown in Figure 1.

Laser polishing is a post-processing technique based on scanning the part’s surface with a laser beam, originally developed to improve the surface quality of metallic parts. This treatment has recently been investigated for polylactic acid (PLA) [11,12,13], fibre-reinforced PLA-FDM [14], and acrylonitrile butadiene styrene (ABS) [12,15,16]. It certainly has more advantages than the traditional polishing techniques, not only thanks to its ability to adapt to the type of material, but also because it is a non-contact polishing technique, which leads to no tool loss. Laser polishing has been successfully used to improve the surface quality of parts fabricated from various materials such as glass, diamond, and various metals [17]. Its operating parameters differ from one material to another. Thus, after parameter optimization on a material, the process can be automated using a robotic arm and can be applied to a wide range of sizes and shapes. Nevertheless, when these beams impact the material surface to be polished, the surface starts to melt owing to the high temperature, especially in the case of polymer materials, which is also addressed by the present study; therefore, an optimization of the laser polishing process parameters is vital in order to obtain optimum results in terms of a reduction in surface roughness along with the clean, unaltered surface of the sample. Among the various additive manufacturing technologies within the current research study, Markforged X7 FDM/CFF 3D printing technology was used to manufacture the tested specimens, using their in-house developed nylon type material, Onyx, which incorporates chopped carbon fibres. This material is purpose-built for the requirements of the aerospace, transportation, and automotive industry, having a strong resistance to solvents and petrochemicals. It is the ideal material for the production of parts that require a nice appearance in compliance with industry requirements. Onyx has been chosen for this work mainly because of its superior resistance to heat compared with other plastic 3D printing materials and because of the varied range of applications in which it could be found. After part-printing, the surface roughness of the 3D-printed samples was measured before and after laser polishing using a MARSURF PS10 surface profilometer.

## 2. Materials and Methods

### 2.1. Methodology Applied for the Experiments in the Present Study

Figure 2 presents the experimental methodology for the laser polishing process developed for improving 3D printed short-carbon fiber reinforced composites surface quality, starting from design to manufacturing and post-processing.

### 2.2. Part Printing

Table 1 summarizes the printing parameters used to develop the specimens. No variation in the printing parameters like layer thickness and part printing orientations was investigated; for all examined Onyx 3D-printed specimens, the layer thickness used was 0.25 mm and the deposition direction was X = 45°, Y = 0°, and Z = 0°.

Onyx can be used alone or reinforced with carbon, glass, or Kevlar. According to Markforged, parts printed with this material exhibit 30% stronger resistance and are stiffer than similar parts made in other 3D printers. Material properties for Onyx reported by the manufacturer are shown in Table 2 [18].

The most important stages in the 3D printing process consist of the following: geometry design definition of the part—3D CAD file, 3D printing parameter definition, and the printing process. As mentioned previously, fused filament fabrication, also known as fused deposition modeling (with the trademarked acronym FDM), has been further developed by Markforged from continuous fibre filament CFF printing. Unlike 3D printing methods like stereolithography (SLA) or digital light processing (DLP), where surface finishing or other intrinsic post-processing operations related to the additive manufacturing technique (e.g., cleaning, UV curing, and so on) are needed, the Markforged X7 printer allows to directly obtain the 3D part. Still, the surface quality requires a significant improvement when specific requirements or applications are targeted. Furthermore, both the surface quality and mechanical properties of 3D-printed parts are affected and governed by pre-processing parameters like printing orientation and layer thickness, as well as reinforcement volume fraction of fibres (if applicable). Most manufacturing processes start from a software model that describes the geometry of the part to be printed. Almost any CAD software can be used and the final model must be a 3D one [19].

The specimens investigated within the present study had a square base prism geometry, shown in Figure 3, with dimensions of 60 mm × 60 mm × 7 mm. The geometry of the specimen was intentionally kept very simple for ease in measurement. Three-dimensional (3D)-printed specimens showed ‘stair-case’ appearance, one of the weaknesses common to all current flat-layer AM technologies, along with lamination weaknesses in a direction perpendicular to the layer direction.

### 2.3. Surface Roughness Measurement

The surface roughness measurements of the 3D-printed samples were performed before and after the laser surface polishing treatment using a MARSURF PS10 surface profilometer. Figure 4 illustrates the fact that three angular positions were used for the measurement (0°—parallel to the 3D printing deposition lines, 90°—perpendicular to the 3D printing deposition lines, and at a 45° angle from the 3D printing deposition lines); therefore, a total of 15 measurements were performed on all five investigated specimens. Table 3 below presents a summary of the surface roughness measurements results obtained before the laser surface polishing treatment.

### 2.4. Surface Finishing Process

Figure 5 shows the experimental setup for the laser surface polishing treatment. The main component of the installation consists of a fibre laser cleaner with a maximum output power of 100 W. The laser radiation is guided through a 2D galvanometer scanner and a focal lens with 255 mm focal length. A PC, command board, RTC^®^ 5 control, and the software interface (laserDESK^®^) form part of the installation.

The laser polishing method was used in this study to enhance the surface roughness of additive manufactured parts. This method is a technique that re-melts to modify the surface morphology without affecting the bulk characteristics. After the samples were 3D-printed, they were exposed to the energy of the fibre laser. Only the surface of the specimens was analyzed in this study. The process consists of the irradiation of the laser onto the 3D-printed surface and determination of the exposed surface to melt. Owing to gravity and surface tension, the liquefied material reorganizes to the same level after the melt pool is generated. After this operation, the temperature of the melted surface drops rapidly, which causes the solidifying phenomenon. The equipment used is presented in Figure 5 and has a P = 100 W maximum power, V = 200 mm/s maximum speed, maximum pulse frequency f = 5–500 kHz, and laser beam diameter of D = 0.03 mm. Any integer percentage of the maximum power, speed, and beam diameter is allowed. Table 4 summarizes the laser source characteristics.

In pulse wave polishing, the energy density (*ED*) formula used for experiments is described below:(1)$$ED=\frac{P}{\frac{\pi \cdot D^{2}}{4}}\times \frac{DC}{f},$$
where *D* is the beam diameter (mm), P represents the pulse power (W), *DC* is the pulse duty cycle (%), and *f* is the pulse frequency (kHz). Note that an excessive energy density resulted in a poor surface quality. Using a more moderate *ED*, laser polishing did not have a noticeable influence upon surface quality.

Prior to defining the optimum parameters presented in Table 5, several tests were conducted to reach the finest range of values for laser power and pulse frequency. It has been found that a bigger frequency (e.g., 90 kHz) resulted in the formation of many debris on the test samples and, by increasing the laser power and minimising the frequency, the results were significantly improved. Note that the scan speed, spot diameter, and line spacing were constant for all experiments. All experiments were performed under nominal environmental conditions (ambient temperature and atmospheric pressure).

## 3. Results and Discussion

### 3.1. Surface Roughness

To determine the reduction in surface roughness, Equation (2) below will be used:(2)$$∆R_{a}=\frac{initial_{roughness}-final_{roughness}}{initial_{roughness}}\times 100$$

As mentioned before, the roughness of the specimens was measured in three directions, namely, parallel, perpendicular (90°), and at a 45° angle with respect to the deposition lines. The same layer deposition angle during 3D printing was used for all investigated specimens (X = 45°, Y = 0°, Z = 0°). Before the laser polishing process, surface roughness measured in the parallel direction for all samples was lower than the roughness measured in the perpendicular direction with respect to the layer deposition lines. Roughness measurements after polishing and reductions in surface roughness for all of the tested specimens are presented in Table 6. Figure 6 presents the roughness measurements at 90° and 45° angles for the investigated specimens following laser polishing post-processing using different energy density (*ED*). No difference between parallel and perpendicular measurements has been noticed after the laser polishing process. Figure 7 shows that the most significant reductions in surface roughness were obtained by specimens 2, 3, and 4. After laser treatment, for specimen 2 (S2) with P = 60 W laser power and f = 40 kHz frequency, the minimum surface roughness value (parallel to laser polishing) obtained was 2.534 µm. Similarly, the minimum surface roughness value (45° angle and perpendicular) was 2.684 µm and 2.601 µm, respectively. As mentioned before, after the part was exposed to the laser energy, the formation of a melt zone (MZ) and heat-affected zone (HAZ) (see Figure 1) annulled the influence of surface roughness measurements according to the directions mentioned before. Therefore, for S3 and S4, the minimum surface roughness values obtained were 1.725 µm and 2.696 µm, respectively (perpendicular to laser polishing). The average areal surface roughness of specimen S3 was reduced from 13.09 µm to 2.06 µm, with a roughness reduction of 84.2%. An important and obvious observation is that different laser polishing parameters have different effects on the surface quality. In the case of specimens S1 and S5, the laser power of P = 60 W and a frequency of f = 90 kHz had a poor effect on enhancing the surface quality because of the debris spreading around the HAZ and solidifying at the same time with the MZ. Then, the optimization came following the idea of reducing the frequency in such s manner that the Onyx particles affected by the polishing process could not influence the area already treated. Therefore, treating smaller surfaces (25 × 20 mm) using the same laser parameters turned out to be more efficient, obtaining a reduction in surface roughness of 71.64%. S1 and S5 were the only two specimens in which only one laser treatment was deployed, thus obtaining greater surface roughness values than specimens S2, S3, and S4, where two successive passages were performed (Figure 6).

The lowest energy density value, achieved by increasing the pulse frequency, reduced the perpendicular surface roughness from 20.167 µm to 7.085 µm (64.86%), which is actually not a significant reduction compared with the results from the other specimens (see Figure 8). Laser polishing began to have a major impact on the surface roughness after a dramatic decrease in pulse frequency occurred, but keeping the laser power in the same range (check Table 4). Therefore, the greatest reduction in this study is achieved by S3 (P = 50W, f = 50 kHz); at an energy density of ED = 1.27 J/mm², a 91.15% reduction in surface roughness is obtained.

### 3.2. Surface Topography

Figure 9, Figure 10, Figure 11 and Figure 12 below present the three-dimensional topography of the S1 and S5 Onyx specimens before and after the laser polishing treatment. A reduction in surface roughness was obtained for all samples, even though not all of them were polished using the optimized parameters. It is observed that, for specimens 1 and 5, the laser polishing direction had a great influence upon the dimensional inaccuracies obtained on the profiles, considering that the same polishing parameters were used for both parts.

Figure 9 and Figure 11 show a comparison of the geometrical profile on polished and original surfaces. In the case of S1, the maximum cross section of the as-printed profile is 0.235 and 0.216 mm in the extremities (Figure 10a) and, for the polished surface, it is 0.627 and 0.575 mm (Figure 10b). The laser polishing path had a major influence on the geometry. It can be seen clearly in the section view of S5 presented in Figure 12 that, by polishing using a 90° angle, the maximum cross section is 1.339 and −0.601 mm, which is a big difference because, for the original surface, the maximum cross section is 0.153 mm.

For specimen S1, a 45° polishing angle was employed. By treating the part in this manner, we obtained a smoother profile, which can be seen in Figure 10b. The red zones represent the surplus material, whereas the blue zones represent the lack of material.

Figure 13 presents the 3D topography of the specimen with the best surface roughness (S3). The maximum cross section (Figure 14b) in the extremities of the polished surface in this case is 0.63 and 0.3 mm. Similar values were obtained for all of the test specimens in which a 45° angle was employed, regardless of the energy density used. It was observed that the largest dimensional deviations were obtained by the 90° angle laser path specimen (S5).

### 3.3. Comparative Study of Process Time and Final Surface Roughness

Finally, to justify the necessity of laser post-processing of the printed parts, a comparative study was conducted of printing time and final surface roughness. In this regard, a test specimen was printed using a layer thickness of 0.05 mm (finest printing available on Markforged X7). The process time of the printed workpiece was measured to be 6 h and 29 min. Surface roughness measurement in three position was conducted. Table 7 summarizes the difference between employing a fine-printing strategy versus laser polishing with respect to the printing time and roughness values.

Even though laser polishing is the winning strategy, dimensional deviations and the laser-affected zone on the subsurface could be of concern. Normally, a reduction in surface roughness is known to reduce the possibility of brittle fracture, but a morphological study is required to evaluate the surface irregularities.

## 4. Conclusions

Generally, post-processing plays an important role in enhancing the mechanical/chemical and esthetical properties, complying to the imposed tolerances. These essential characteristics of the surfaces lead to the increase in demand at the industrial level of the post-processing operations. Even though this study is focused on the change in surface roughness, a comprehensive experimental study on the mechanical properties of the treated part should also be taken in consideration. Therefore, laser polishing was employed to improve the surface quality of Onyx specimens. Modification in surface roughness and geometrical deviations was studied. The main conclusions are as follows:After laser polishing with the optimal parameters, the surface roughness of Onyx specimens was greatly reduced, especially for S2, S3, and S4. Sample S3 obtained the greatest reduction from 19.5 µm to 1.725 µm.The increase in the number of passes and reducing laser power and frequency improves the smoothing.For higher laser power and frequency values, the geometry of the test samples was greatly affected. Therefore, Onyx reacted better at a lower frequency and no debris was found on the test samples.Using the same laser parameters for S1 and S5, by changing the laser path, it has been observed that it had little influence on the surface roughness, but had a major impact on the geometrical deviation of the samples.While the laser process involves material removal/melting, causing a dimension variation, an experimental study on the mechanical properties of the treated part should also be taken in consideration.

In this work, an approach to improve the surface quality of new additively manufactured polymer parts using the laser polishing process was developed. These results can be of high interest in the context of intensive interest for the use of newly developed continuous filament fabrication (CFF) 3D printing technology, as an alternative solution for conventional CFRTPC composites developed using techniques like vacuum forming, filament winding, pultrusion, bladder-assisted moulding, or compression. These conventional processes require expensive facilities and equipment, such as autoclaves or complex rigid molds for out-of-autoclave processes, hindering the wide application of composites.

## Figures and Tables

**Figure 1 materials-15-07398-f001:**
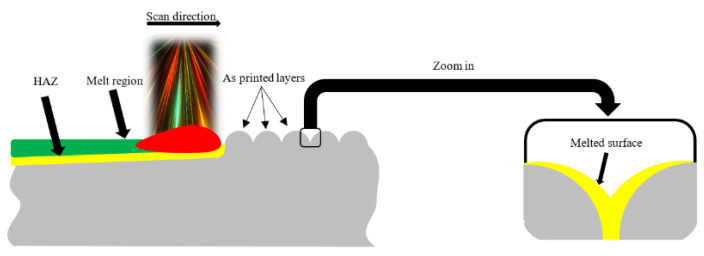
The laser polishing process.

**Figure 2 materials-15-07398-f002:**
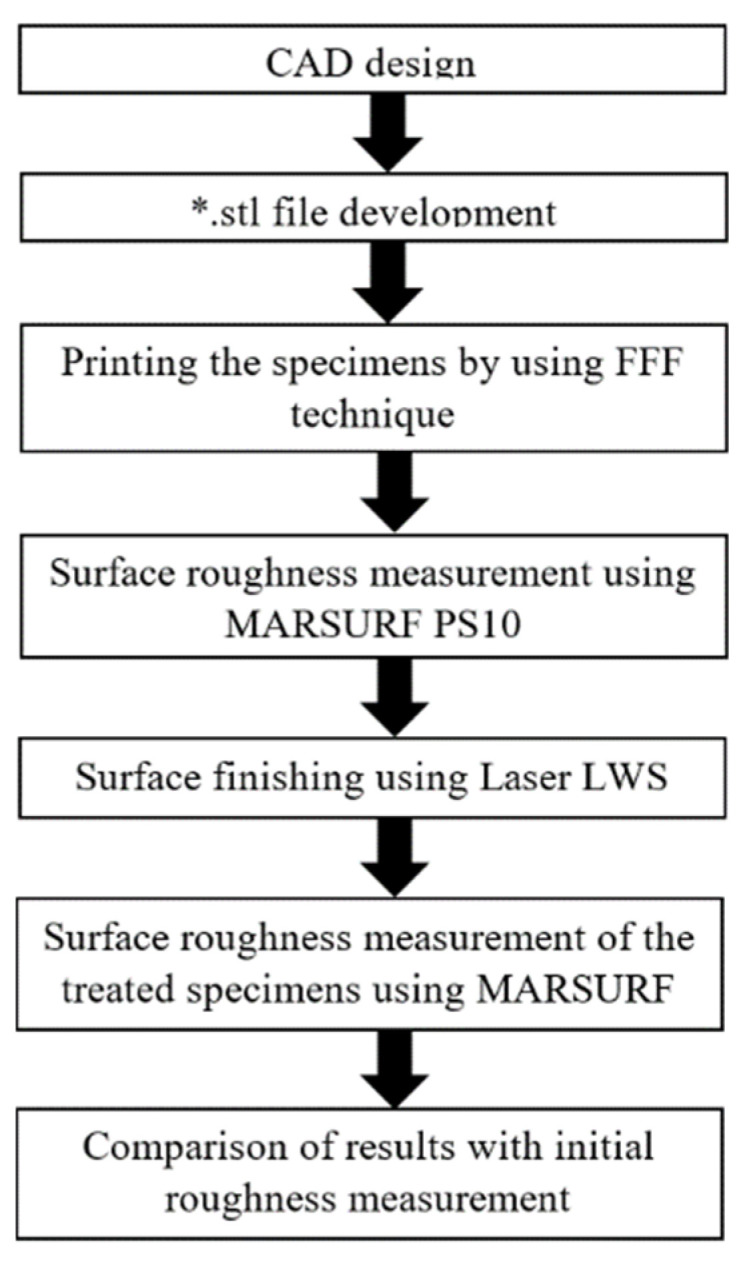
Experimental methodology from design to manufacturing and post-processing.

**Figure 3 materials-15-07398-f003:**
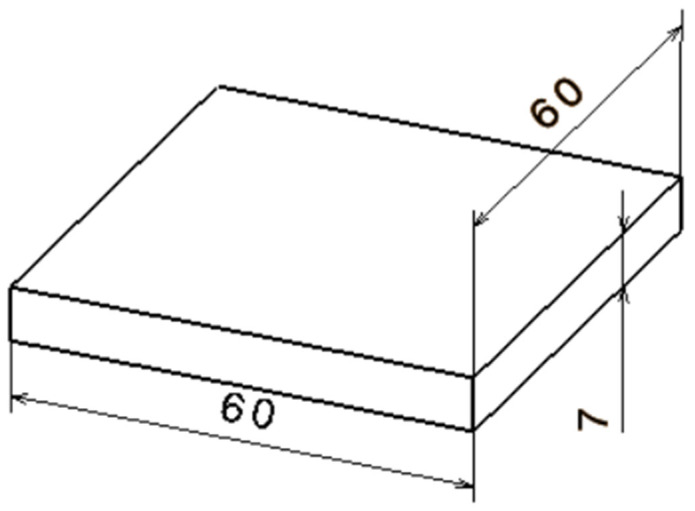
Onyx 3D-printed specimen sketch.

**Figure 4 materials-15-07398-f004:**
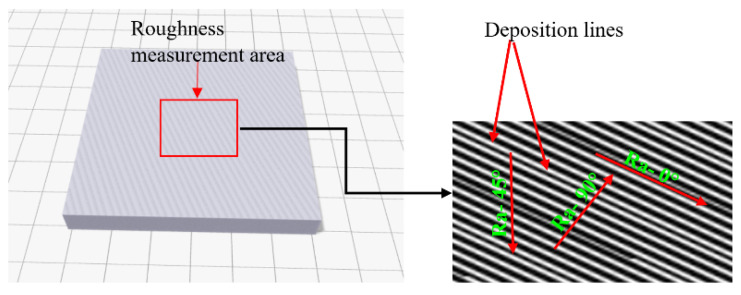
Roughness measurement strategy.

**Figure 5 materials-15-07398-f005:**
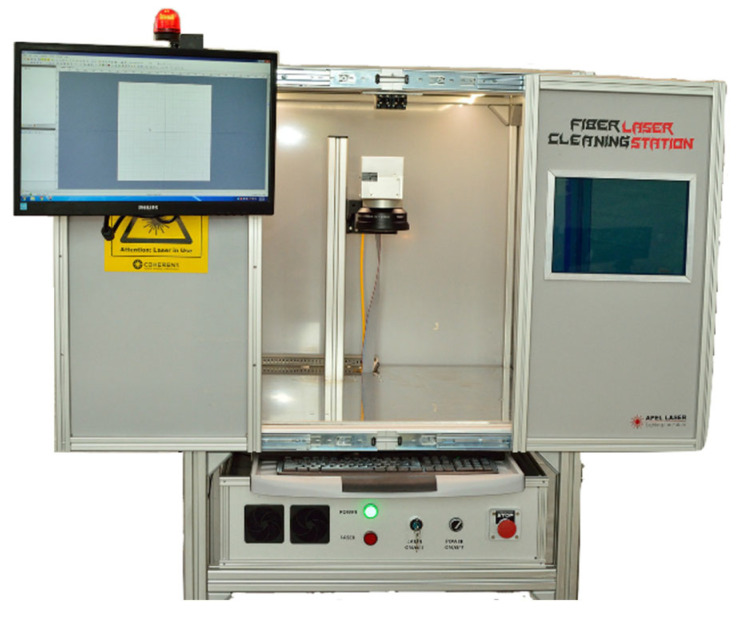
Experimental setup for the laser surface polishing treatment.

**Figure 6 materials-15-07398-f006:**
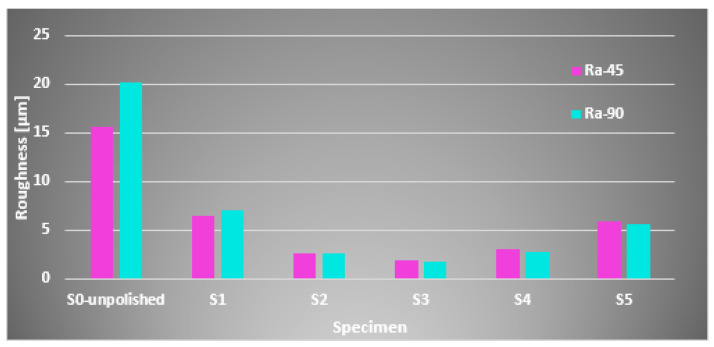
Roughness measurements at 90° and 45° angles for the investigated specimens following laser polishing post-processing using different ED.

**Figure 7 materials-15-07398-f007:**
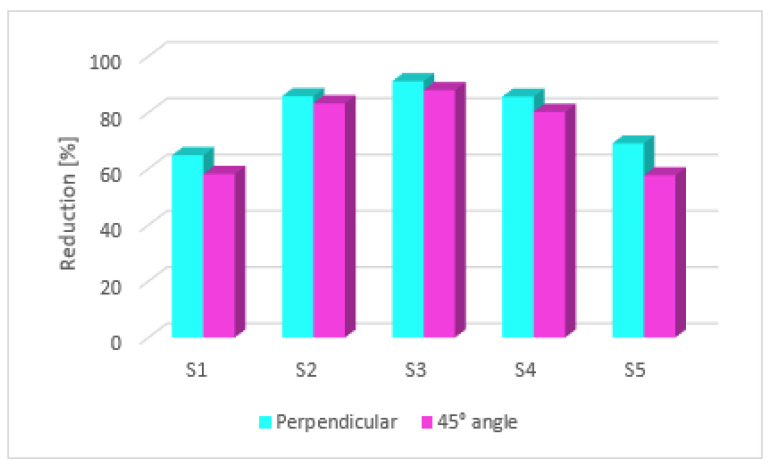
Roughness reduction at 90° and 45° angle after laser polishing post-processing.

**Figure 8 materials-15-07398-f008:**
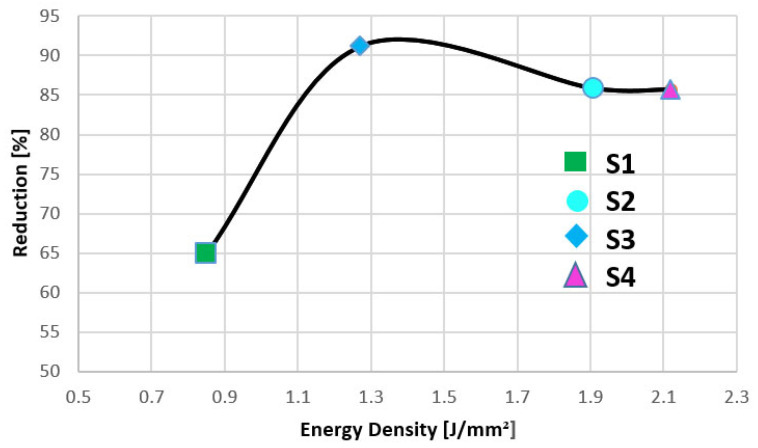
Variation in surface roughness reduction (measured at 90°) with energy density.

**Figure 9 materials-15-07398-f009:**
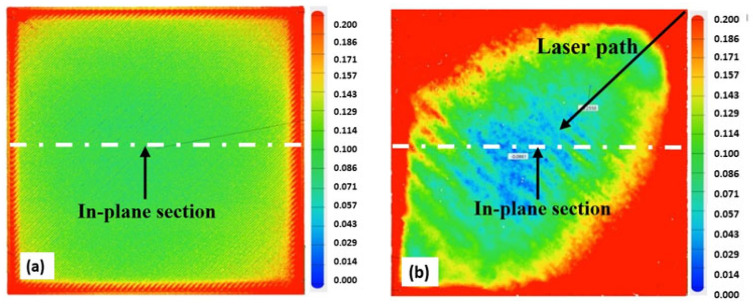
Three-dimensional (3D) topography of the specimen before and after the laser polishing: (**a**) unpolished; (**b**) S1: 60 W laser power, 90 kHz frequency.

**Figure 10 materials-15-07398-f010:**
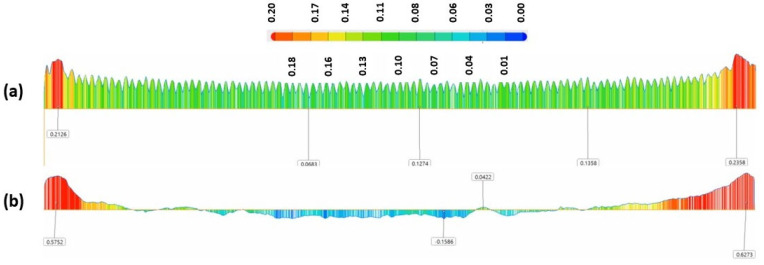
In-plane section view of S1; (**a**) before and (**b**) after laser polishing.

**Figure 11 materials-15-07398-f011:**
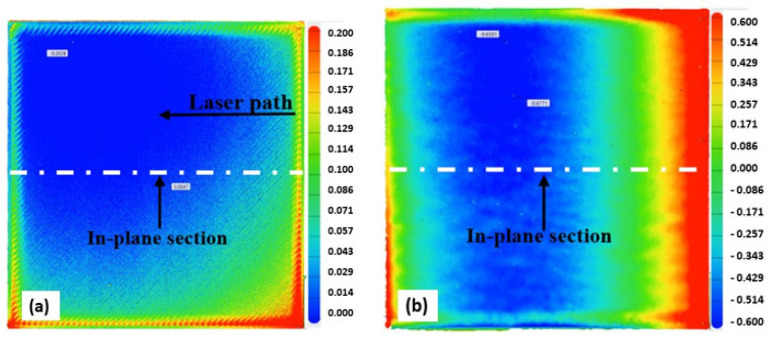
Three-dimensional (3D) topography of the specimen before and after the laser polishing: (**a**) unpolished; (**b**) S5: 60 W laser power, 90 kHz frequency.

**Figure 12 materials-15-07398-f012:**
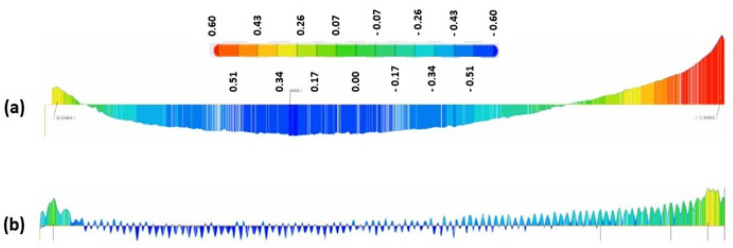
In-plane section view of S5; (**a**) before and (**b**) after laser polishing.

**Figure 13 materials-15-07398-f013:**
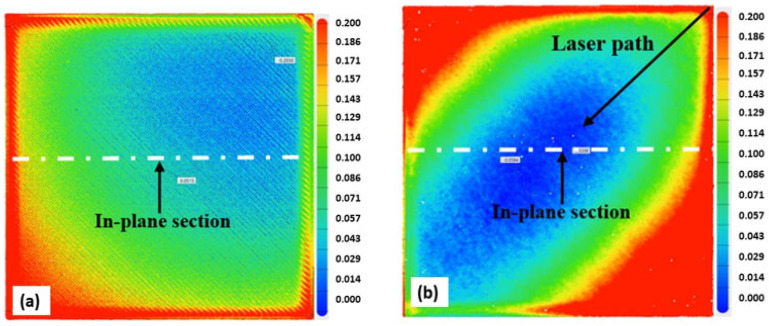
Three-dimensional (3D) topography of the specimen before and after the laser polishing: (**a**) unpolished; (**b**) S3: 50 W laser power, 50 kHz frequency.

**Figure 14 materials-15-07398-f014:**
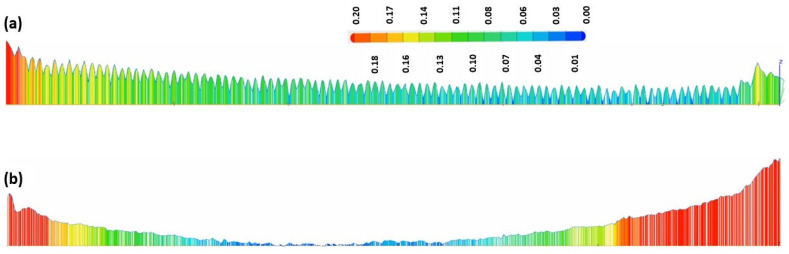
In-plane section view of S3; (**a**) before and (**b**) after laser polishing.

**Table 1 materials-15-07398-t001:** FFF 3D printing parameters of Onyx specimens.

Material	Onyx
Nozzle diameter (mm)	0.4
Nozzle extrusion temperature (°C)	270
Heat bed temperature (°C)	N/A
Deposition layer height (mm)	0.25
Deposition layer width (mm)	0.4

**Table 2 materials-15-07398-t002:** Mechanical properties of Onyx.

Young’s modulus [GPa]	1.4
Yield stress [MPa]	36
Ultimate stress [MPa]	30
Flexural strength [MPa]	81
Flexural modulus [MPa]	2.9
Density [$g/{\mathrm{cm}}^{3}$]	1.2

**Table 3 materials-15-07398-t003:** Surface roughness values before laser polishing.

Sample	Roughness $R_{a}$ (µm)		
	0°	45°	90°
S1	4.715	15.581	20.167
S2	4.691	15.183	18.349
S3	4.478	15.284	19.508
S4	4.745	15.417	18.834
S5	3.932	13.919	18.262

**Table 4 materials-15-07398-t004:** Laser source characteristics.

Characteristics	Unit	Value
Source type	-	Fibre laser
Wavelength	nm	1064
Maximum power	W	100
Focal length	mm	255
Working area	${\mathrm{mm}}^{2}$	170 × 170
Pulse frequency	kHz	5–500
Pulse width	ns	80–120
Working temperature	°C	25 ± 10 °C

**Table 5 materials-15-07398-t005:** Laser surface polishing treatment parameters used in this study.

Parameters	Specimen				
	S1	S2	S3	S4	S5
Pulse power (W)	60	60	50	50	60
Pulse frequency (kHz)	90	40	50	30	90
Scan speed (m/s)	1	1	1	1	1
Spot diameter (mm)	0.03	0.03	0.03	0.03	0.03
Line spacing (mm)	0.05	0.05	0.05	0.05	0.05

**Table 6 materials-15-07398-t006:** Roughness measurements after polishing and reductions compared with the initial measured roughness values.

**Sample**	Energy Density $\left(J/{\mathrm{mm}}^{2}\right)$	Roughness $R_{a}$ (µm)			Reduction %		
		0°	45°	90°	0°	45°	90°
S1	0.85	8.142	6.513	7.085	−72.68	58.199	64.868
S2	1.91	2.684	2.534	2.601	42.784	83.310	85.824
S3	1.27	2.641	1.834	1.725	41.022	88.000	91.157
S4	2.12	2.777	3.040	2.696	41.475	80.281	85.685
S5	0.85	5.738	5.888	5.651	−45.93	57.698	69.055

**Table 7 materials-15-07398-t007:** Difference between the fine 3D printing strategy and laser polishing post-processing solution (ED = 1.27 J/mm²).

Process	Process Time [hh:mm]	Average Ra [µm]
Fine printing	06:29	5.26
Turbo printing + post-processing	01:01	1.72
**Reduction [%]**	**84**	**67.3**

## Data Availability

Not applicable.

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
