# Peer review of "Reducing Surface Roughness of 3D Printed Short-Carbon Fiber Reinforced Composites"

_materials, 2022, doi:10.3390/ma15207398_

Round 1

Reviewer 1 Report

Fused deposition modelling (FDM) additive manufacturing is an effective technology for 3D workpieces of arbitrary shape. Laser polishing is an effective technique for reducing the surface roughness of the materials. Here, the authors have demonstrated the effect of laser polishing on the surface roughness of fused deposition-modeled components. The paper can be accepted after a miner revision

1.     Authors need to recheck all the Figures to make the text clearer, especially Figures 10-13.

2.     Why using line roughness to describe the surface quality rather than surface roughness.

3.     The pulse width of the laser should be provided in the Laser source characteristics section.

Reviewer 2 Report

Manuscript review „Reducing surface roughness of 3D printed Onyx parts”.

The manuscript is very interesting, but requires substantive and editorial correction.

The summary needs to be edited again. The abstract describes the content of the manuscript. parts of the text (7-17) should not be included in the summary.

The content in the abstract should be placed in the introduction

(68-84) Re-editing of the text is required.

(102; 154; 166; 260; 263; and next) - the quality of the drawings should be improved.

There is no information on the mechanical and technological properties of samples made in different technologies.

The final quality of the product printed in 3D printing technology also depends on the use of post-printing processes - were there any processes and possibly what processes?

There is no information on the environmental conditions under which the tests were conducted ??

Conclusions very sketchy and require extensive discussion undertaken research problem.

Reviewer 3 Report

In this paper the authors attempted to reduce the surface roughness of 3D printed Onyx parts. They have studied the influence of different parameters on surface roughness and dimensional accuracy, such as: laser power, pulse frequency, laser scanning path.

I have some comments on this work:

1. Please describe the novelty of your work.

2. Please explain why did you chose the Onyx as an experimental material. Where does this material usually applied in the industry? What are the main problems while manufacturing this material using AM.

3. The literature review is not sufficient. Please provide more studies performed in this filed related to the 3D printing of Onyx. Specially about the influence of main laser process parameters on the surface roughness and dimensional accuracy. Sufficient information about the previous studies should be added and compared with the current work so that the reader could follow the present study rational.

4. Please explain about the design of your experiment. Which method was used in your paper?

5. Nothing is mentioned about Table 1. Please explain about Table 1 in the text.

6. Section 2.4., line 169. I think "Figure 4 shows ....." should be replaced by "Figure 5 shows .....". Please check it.

7. Page 7, line 208: I think "As shown in Figure 4  ....." should be replaced by "As shown in Figure 6  .....". Please check it.

8. Please provide Figures 9 to 14 with better qualification. It is very difficult to read the numbers on these pictures.

Round 2

Reviewer 3 Report

The authors have answered clearly to the all questions. So, the paper can be published as it is.
